# Evaluating the Effects of an Enhanced Strength Training Program in Remote Cardiological Rehabilitation: A Shift from Aerobic Dominance—A Pilot Randomized Controlled Trial

**DOI:** 10.3390/jcm13051445

**Published:** 2024-03-01

**Authors:** Irene Nabutovsky, Roy Sabah, Merav Moreno, Yoram Epstein, Robert Klempfner, Mickey Scheinowitz

**Affiliations:** 1Faculty of Medicine, Tel Aviv University, Tel Aviv 6997801, Israel; robert.klempfner@sheba.health.gov.il; 2Cardiac Prevention and Rehabilitation Institute, Leviev Heart Center, Sheba Medical Center, Ramat Gan 5266202, Israel; merav.moreno@sheba.health.gov.il; 3School of Public Health, Faculty of Medicine, Tel Aviv University, Tel Aviv 6997801, Israelyoram.epstein@sheba.health.gov.il (Y.E.); 4Department of Biomedical Engineering, Faculty of Engineering, Tel Aviv University, Tel Aviv 6997801, Israel; mickeys@tauex.tau.ac.il; 5Sylvan Adams Sports Institute, School of Public Health, Faculty of Medicine, Tel Aviv University, Tel Aviv 6423906, Israel; 6Neufeld Cardiac Research Institute, Sheba Medical Center, Ramat Gan 5266202, Israel

**Keywords:** cardiac rehabilitation, resistance training, muscle endurance, remote health monitoring, patient engagement

## Abstract

(1) **Background:** Cardiac rehabilitation often emphasizes aerobic capacity while overlooking the importance of muscle strength. This study evaluated the impact of an enhanced remote strength training program (RCR-ST) on cardiac rehabilitation. (2) **Methods:** In this randomized prospective study (RCT registration number SMC-9080-22), 50 patients starting cardiac rehabilitation were assessed for muscle strength, aerobic capacity, and self-reported outcomes at baseline and after 16 weeks. Participants were divided into two groups: the RCR-ST group received a targeted resistance training program via a mobile app and smartwatch, while the control group received standard care with general resistance training advice. (3) **Results:** The RCR-ST group demonstrated significant improvements in muscle endurance, notably in leg extension and chest press exercises, with increases of 92% compared to 25% and 92% compared to 13% in the control group, respectively. Functional assessments (5-STS and TUG tests) also showed marked improvements in agility, coordination, and balance. Both groups improved in cardiorespiratory fitness, similarly. The RCR-ST group reported enhanced physical health and showed increased engagement, as evidenced by more frequent use of the mobile app and longer participation in the rehabilitation program (*p* < 0.05). (4) **Conclusions:** Incorporating a focused strength training regimen in remote cardiac rehabilitation significantly improves muscle endurance and patient engagement. The RCR-ST program presents a promising approach for optimizing patient outcomes by addressing a crucial gap in traditional rehabilitation protocols that primarily focus on aerobic training.

## 1. Introduction

Cardiac rehabilitation represents a critical phase in the continuum of care for patients with cardiovascular diseases, aiming to enhance their physical and psychological well-being, reduce the risk of recurrent events, and improve their overall quality of life [1]. Traditional cardiac rehabilitation (CR) programs have predominantly centered on aerobic exercises, such as walking, cycling, and swimming [2]. While these interventions have proven beneficial, they may not fully address the multifaceted needs of this patient population, particularly in terms of muscle strength development [3].

Remote cardiac rehabilitation (RCR) has gained prominence in recent years as an effective means of enhancing physical work capacity, patient engagement, and overall satisfaction during the rehabilitation process [4,5]. Previous studies have consistently demonstrated its efficacy, often assessing improvements in physical work capacity through stress tests, yet with a predominant focus on aerobic exercises [6]. The traditional approach to RCR frequently quantified patient progress based on the cumulative minutes of aerobic activity per week, leaving strength training largely overlooked or underperformed [7,8].

The purpose of this study was to evaluate the efficacy of a novel RCR program that, in response to the unmet need described above, prioritizes the development of muscle strength by employing digital applications and remote monitoring to create a more personalized approach. This enhanced program presents an alternative to the prevailing remote rehabilitation paradigms, which tend to be heavily reliant on aerobic activities and often lack an effective strategy for encouraging and monitoring strength training compliance.

To rigorously evaluate the efficacy of this innovative approach, we conducted a controlled randomized pilot trial involving low-risk cardiology patients undergoing RCR at the Sheba Medical Center. This intervention program takes a distinctive stance by centering on strength training while retaining the core objectives of traditional remote rehabilitation. Under the guidance of a physiologist, each patient was tasked with adhering to a weekly, progressively structured strength training protocol designed to address individual needs and capabilities.

This study represents a significant shift in the landscape of cardiac rehabilitation, recognizing the importance of muscle strength development as a fundamental component of overall recovery. By introducing a novel approach that integrates digital tools, individualized strength training, and remote monitoring, our research aims to address the limitations of existing remote rehabilitation programs [2,9], providing a more comprehensive and effective rehabilitation experience for patients on their journey to cardiovascular health. The findings of this study offer promising insights into the potential transformation of remote cardiac rehabilitation paradigms, with the ultimate goal of improving the life and well-being of patients in need.

## 2. Materials and Methods

This was a single-center, randomized-control double-arm, prospective study (RCT registration number SMC-9080-22 13 June 2022, Sheba medical center, Ramat Gan, Israel) in which all patients who met the inclusion criteria participated. The study protocol was reviewed and approved by the Ethical Review Board of the Sheba Medical Center. This study was conducted to evaluate the effects of a remote cardiac rehabilitation program focused on strength training (RCR-ST) on muscle endurance, functional capabilities, cardiorespiratory fitness, and adherence, compliance, and mental health among low-risk cardiac patients. The study cohort was divided into two groups: the intervention—RCR-ST group and the control RCR group, which underwent the standard remote cardiac rehabilitation care. Figure 1 represents the flow diagram of the study.

The study inclusion criteria were based on national guidelines and are accepted indications for CR. The principal inclusion criteria included a left ventricular fraction ≥ 50%. Exclusion criteria included: severe orthopedic, neurological, or cognitive impairment, clinical ischemia, noninvasive evidence of ischemia, diagnosis of heart failure, or low functional capacity, defined as <4 METs at baseline stress test or changed disease state during the program.

The primary endpoint was the change in muscle endurance (the absolute number of repetitions, as well as a percentage change compared to the starting point) demonstrated by patients in leg extension and inclined chest press tests.

Uniform instructions were given and memorized by all participants to ensure consistency in the test conditions. Participants were instructed to execute the maximally possible repetitions of the complete range of movements while exerting the utmost effort, utilizing all resources at their disposal. All evaluations were conducted by the same examiner.

### 2.1. The Muscle Endurance Tests

Leg extension test: This test evaluated the endurance of lower body muscles while predicting performance and functional independence in adults [10]. The inability to perform this exercise with 30% of body weight was indicative of significant functional weakness [11]. Participants were seated on a specialized knee flexion simulator, maintaining a joint angle of approximately 70 degrees, measured using the Goniometer application on an iPhone device, which was pre-validated for reliability [12].

Chest press test: To assess upper body muscular endurance, the chest press test was performed on an inclined bench [13,14]. Successful performance of this test was noted to be strongly correlated (r = 0.8) to upper body muscle function [15]. Participants were required to perform maximal repetitions with resistance set to 30/40% of their body weight (40% for men, 30% for women). Participants were seated on a specialized simulator on an inclined bench at a 30° angle, with their feet positioned on a step. During the exercise, the participants pressed the simulator handles until their elbows were fully extended. Repetitions were performed slowly, maintaining control over the rhythm of all exercises, although the speed of each repetition was not quantified.

Grip strength test: This test assessed the maximum isometric strength of the hand and forearm muscles [16]. Participants exert maximal isometric force by squeezing the dynamometer. Three measurements were taken from each subject, with a one-minute break between each attempt. The final score was determined as the average of the two best results.

### 2.2. The Secondary Endpoints Were

(1)Functional capability changes were evaluated using functional assessments (explanation below).(2)Cardiorespiratory fitness was evaluated by defining metabolic equivalents (METs) using a stress test on treadmill with the Bruce Protocol [17]. The assessments were conducted by the patient care team at the rehabilitation center, with team members kept unaware of the participants’ affiliation with the research groups.Assessments were conducted at the commencement of the rehabilitation program and subsequently after a 4-month intervention period.(3)Compliance and adherence: The following variables were evaluated longitudinally during each week of the program: the total number of minutes of aerobic exercise (aerobic minutes), the number of aerobic minutes in the target heart rate, the assessment of perceived Borg scale, the number of training sessions, the number of daily steps, the use of the RCR mobile app (number of weekly entries). The duration of participation in the CR program (number of weeks) was also noted.(4)Questionnaires defining mental and physical health (PROMISE 10) [18] and PHQ9 [19] were evaluated. These questionnaires were sent out using the app at enrollment and after 4 months of intervention.

### 2.3. Functional Assessments

In these assessments, the duration of the entire exercise is measured in seconds, with all tests administered by the same examiner.

Five-time sit–stand test (5STS): Seated participants were required to stand up and sit down five times as quickly as possible without using their hands. This test has been validated to predict the functional performance of the individual, and it reflects their physical condition [10,20].

Timed “Up and Go” test (TUG): Patients were instructed to stand up, walk three meters in a straight line, turn around, return to the chair, and sit down. A favorable outcome in this test is associated with high functional independence, while an unfavorable result is linked to functional decline and an increased risk of falls [21].

### 2.4. The Intervention Program

The intervention program was based on a successful existent RCR program for low-risk patients, which has been conducted at Sheba Hospital in Israel since 2018 and has been described in detail in previous papers [8,22,23]. Low-risk status according to the national guidelines is defined as a lack of >5% ischemia per stress or pharmacological single-photon emission computed tomography (SPECT), left ventricular ejection fraction (LVEF) of 50% or more, a lack of sustained ventricular tachycardia, and symptomatic atrial fibrillation of flutter or heart rates (HRs) > 120 beats per minute at rest. The low-risk definition excludes patients following cardiac arrest or heart failure with preserved systolic function and low functional capacity (metabolic equivalents (METs) < 6).

Briefly, the traditional RCR program combined the use of advanced telecare technologies (platform by Datos Health, Tel Aviv, Israel) and smart wearable devices (mostly Garmin and Apple smartwatches) (Figure A1 and Figure A2, Appendix A) adapted to the characteristics and needs of each patient. The program was geared to motivate the patient to perform workouts at home or in any convenient community facility, to lead a healthy lifestyle, as well as manage their clinical and psychological health. Both motivational and technological conditions are needed for RCR success. For example, in addition to receiving automatic encouraging messages and reminders, the patient received a personal weekly call from a rehab trainer to discuss the training program and any problems that may have arisen. The patient was given necessary technical and clinical support from a multiprofessional care team (physician, psychologist, physiologist, nutritionist, and nurse).

The established recommendations within the RCR program entail engaging in 150 min of weekly aerobic exercise, with a minimum of 120 min within the target heart rate range. Additionally, participants are advised to undertake two strength training sessions weekly and achieve a daily step count of 8000. To facilitate strength training, patients are furnished with video clips illustrating diverse strength exercises, which are automatically distributed through the app.

The current intervention has placed a special emphasis on strength training, in addition to all the usual goals of the RCR program. The patients received additional instructions and a program with eight exercises for the main muscle groups (Figure A3), which they had to perform regularly 2–3 times a week, starting from the third week of rehabilitation.

The program was provided to patients through a digital brochure and the app’s video files, with detailed explanations. Each exercise had three levels of difficulty. The 16-week program included 2 weeks of physical preparation (aerobic training only) and familiarization with the work of the app and smartwatch and a further 14 weeks of strength exercises, gradually increasing the number of workouts, sets, and complexity level.

The progress in training was determined every week by the rehabilitation trainer in collaboration with the patient, commensurate with the patient’s performance and success. Also, patients received special educational content that emphasized the importance of strength training for better rehabilitation and motivated the patient to better compliance.

The control group underwent a standard RCR program for 16 weeks that included general recommendations for performing strength training twice a week, starting from the 5th week of rehabilitation. Participants were also provided with educational materials, video content, and an exercise brochure and engaged in weekly discussions with a trainer, with a primary emphasis on aerobic training.

### 2.5. Statistical Analysis

Descriptive demographic and clinical characteristics are presented as frequencies and percentages for categorical variables and as mean ± standard deviation (SD) for continuous variables. Chi-square statistics or student’s *t*-test were used as appropriate for comparison of groups of patients as appropriate. Two-way repeated measures ANOVA tests were conducted to examine the effects of experimental intervention on outcome measures at baseline and after 16 weeks. Pearson coefficient correlations were used to identify the relationships between continuous variables. All tests were two-sided, with a *p* value of less than 0.05 considered significant. Statistical analyses were performed using SPSS, version 27.0 (SPSS Inc., Chicago, IL, USA).

## 3. Results

A total of 50 participants were recruited and randomized: 23 in the intervention group (15 completed the 16-week RCR-ST program) and 27 in the control group (21 completed the 16-week follow-up). Demographics and clinical characteristics are presented in Table 1. (Demographics and clinical characteristics of the groups included in the final analysis are presented in Appendix A Table A1). The majority of participants were middle-aged (59.8 ± 10.4), male (88%), and non-smokers (90%). The most frequent comorbidities were dyslipidemia (32%) and hypertension (30%). The main indications for CR were ischemic heart disease (80%), percutaneous coronary interventions (58%), and myocardial infarction (58%). Baseline characteristics were similar between the two groups, except for male predominance in the control group (96.3% vs. 78.3%, *p* = 0.050).

### 3.1. Analysis of Muscle Endurance

Significant interaction effects between time and group were observed for the number of leg extensions (*F*(1,34) = 9.02, *p* = 0.005, η2 = 0.210) and chest press repetitions (*F*(1,34) = 28.49, *p* < 0.001, η2 = 0.456). After 16 weeks of intervention, the RCR-ST group showed significantly greater improvement in both measures compared with the control group: delta 14.3 ± 8.6 vs. 4.8 ± 9.8 for leg extensions and delta 9.7 ± 6.1 vs. 1.7 ± 2.8 for chest press (Table 2). However, no significant interaction effect of time and group was found on grip strength (*F* < 1).

### 3.2. Analysis of the Functional Assessments 

Significant interaction effects between time and group were detected for the 5STS Test (*F*(1,34) = 5.91, *p* = 0.021, η2 = 0.148) and TUG Test (*F*(1,34) = 6.08, *p* = 0.019, η2 = 0.152). After completing the program, the RCR-ST group showed significantly greater improvement as reflected by faster performance times in tests compared with the control group: delta 2.7 ± 1.9 vs. −1.1 ± 1.9 for 5STS and delta −2.7 ± 3.6 vs. −0.5 ± 1.6 for TUG (Table 2).

### 3.3. Analysis of the Secondary Outcomes

A significant main effect was observed on METs (*F*(1,31) = 44.74, *p* < 0.001, η2 = 0.591), indicating that the scores were significantly higher after 16 weeks than at baseline for both groups. However, the interaction effect between time and group was not significant (*F*(1,31) = 2.36, *p* = 0.134, η2 = 0.071).

Furthermore, a significant interaction effect between time and group was found in physical health functioning, as measured by PROMIS-10 (*F*(1,23) = 4.27, *p* = 0.050, η2 = 0.156). After a 16-week intervention, the RCR-ST group showed significantly greater improvement than the control group: delta 4.6 ± 3.5 vs. −0.1 ± 7.1 (Table 2).

No significant effects of time and group were found in mental health functioning, as measured using PROMIS-10 (*F*(1,23) = 4.27, *p* = 0.050, η2 = 0.156) and by PHQ9 (*F* < 1). Figure 2 provides a summary of the percentage change differences between the groups in the main outcome measures.

### 3.4. Analysis of Compliance and Adherence

The average minutes per week that participants performed aerobic exercise and the average minutes that they spent in the target heart rate range (or above) did not differ between the two groups. However, the control group participants spent more time exercising below the target heart rate range than the intervention group. The intervention group reported performing more resistance training sessions per week than the control group. The number of weekly entries to the mobile application and the duration of participation in the CR program were significantly greater in the intervention group (Table 3). No significant differences between the two groups were observed in the average perceived exertion reported using the Borg scale, the number of aerobic training sessions, and the number of daily steps.

Correlation analysis (Table 4) revealed that muscle endurance, measured using leg extensions and chest press, was positively associated with the intensity of aerobic activity (METs) and physical health functioning (PROMIS-10) and negatively associated with depression scores in PHQ9. Both muscle endurance measures were negatively related to TUG, whereas only the leg extension measures were negatively related to 5STS, indicating that greater muscle endurance was accompanied by faster performance times in functional tests. Grip strength was not significantly correlated with any variables.

## 4. Discussion

### 4.1. Effectiveness of the Enhanced Strength Training Program

This study aimed to address a critical gap in existing distance rehabilitation programs, which tended to focus on aerobic exercise with a limited focus on strength training [2,6,7] In our study, we presented an intervention program specifically aimed at developing muscle endurance in low-risk cardiac patients undergoing remote rehabilitation.

The implementation of a systematic and gradual strength training protocol significantly contributed to improvements in muscle endurance, as evidenced by various physical performance assessments. Patients in the intervention group demonstrated greater advancements in muscle endurance exercises, such as knee extension and chest press, compared to the control group. Although both groups improved at the final testing, the intervention group’s substantial increase in repetitions and percentage improvement in both exercises serves as a reliable marker of enhanced muscle endurance.

The results indicate that a shift from predominantly aerobic training to a more contemporary approach incorporating strength exercises alongside aerobic activity yields positive outcomes in patients undergoing rehabilitation. Based on numerous previous studies emphasizing the importance of muscle strength and endurance for overall health and rehabilitation [24,25,26,27], especially in older patients [27,28,29], our program represents a significant move toward a holistic cardiac rehabilitation strategy.

Furthermore, the intervention group exhibited more significant enhancements in functional abilities, as reflected by the 5STS and TUG tests, which assess agility, coordination, and balance—critical aspects for older individuals. These improvements indicate better performance of the main muscle groups essential for human function and movement. Recognizing the importance of these fitness characteristics, strength training has emerged as an essential tool for improving these key functional aspects. Our findings reveal that the strength training program significantly aided the intervention group in improving their performance in agility, coordination, and balance compared to the control group. The observed reduction in the time required to complete basic motor tasks, such as walking, sitting, standing up, and turning, indicates the strengthening of the central muscle groups responsible for movement and daily body function [30,31,32]. This result underscores the effectiveness of strength training in enhancing the quality of life for this population.

Regarding patients’ aerobic capacity, we did not observe a distinct advantage of the strength program over the conventional one. These findings align with existing literature on the undeniable benefits of both traditional and distance rehabilitation programs for improving aerobic performance [33,34,35]. The relationship between strength training and aerobic exercise is still a matter of ongoing debate. On one hand, it could be logically assumed that intensive strength training might adversely affect the adherence to aerobic training due to constraints on time resources. Additionally, certain studies have proposed that the concurrent inclusion of strength and aerobic training sessions in the same program may compromise the effectiveness of muscle mass impact [36], while other research has indicated a positive influence of strength exercises on aerobic capacities [37,38,39]. Notably, this study did not substantiate either of these assertions. Patients in the intervention group trained as frequently as, if not more often than, the control group, with no adverse impact on their stress test scores, which were comparable to those of the control group.

In accordance with existing literature [40,41,42], this study validated a significant correlation between the muscular and aerobic capacities of patients. Each test assessing muscle endurance or functionality exhibited a consistent correlation with the METS level in the stress test. The superior indicators of muscular endurance and functional abilities were consistently linked to enhanced aerobic capacities. This pattern persisted in both groups and was observed at both the outset and conclusion of the intervention.

### 4.2. Patient-Reported Outcomes

Regular physical activity has consistently demonstrated a positive impact on individuals’ mental health in numerous studies [43,44,45], manifesting in significant reductions in symptoms of depression and anxiety [46,47,48]. However, in this study, the training did not yield a notable effect on the subjective perception of mental health or the assessment of depression in the patients. This may be attributed to the fact that the study cohort consisted of low-risk patients who, by and large, did not have preexisting mental health disorders. Additionally, participants in rehabilitation programs are typically a more motivated and mentally balanced group, while individuals with more complex health issues often do not seek assistance at rehabilitation centers.

Nevertheless, our program exhibited a favorable impact on the subjective evaluation of physical health; individuals in the intervention group rated themselves as feeling physically healthier. This cautiously suggests that a more comprehensive rehabilitation program, incorporating strength exercises, contributes to a more positive self-perception regarding body and physical capabilities [49,50]. Subsequent analysis indirectly supports this assumption, revealing a correlation between higher subjective assessments of physical health and improved muscular endurance, better functionality, and higher METS levels. Remarkably, these positive outcomes were also associated with lower levels of depression.

### 4.3. Patient Compliance and Adherence

A crucial objective of our study was to identify a more effective approach conducive to improved adherence to cardiac rehabilitation goals. Our prior research [51,52], alongside other studies [53], has indicated that remote rehabilitation programs successfully achieve compliance with aerobic objectives but consistently fall short in terms of adherence to prescribed strength training.

A notable achievement of our study is its ability to overcome the longstanding issue of patients not adhering to strength training protocols during cardiac rehabilitation. As anticipated, the intervention group displayed a high level of commitment to strength training, which was likely attributable to the structured program design, individualized progressive plans, and the integration of digital monitoring and support applications.

Unexpectedly, patients in the RCR-ST group also exhibited superior performance in aerobic training. Contrary to our initial expectation that focus on the strength program might compromise aerobic training, these patients not only excelled in strength training but also met other rehabilitation goals, including total daily steps, weekly aerobic duration, and time within the target heart rate.

Furthermore, unproductive training time significantly decreased in the RCR-ST group, indicating more effective and intensive workouts. An increase in muscle mass might have contributed to this improvement [54,55,56]. Additionally, the intervention group demonstrated notably higher engagement in tracking their workouts, utilizing the app more frequently than the control group.

Ultimately, the intervention group displayed markedly better results in evaluating the overall duration of the rehabilitation program, with patients remaining enrolled for a more extended period compared to the control group. These findings suggest that the integration of technology and personalized exercise plans can substantially enhance patient engagement, motivating them to better achieve program goals and fostering prolonged program participation.

At the same time, it is pertinent to note that despite the success of the intervention, regrettably, a significant dropout occurred in both study groups. This lamentable situation in the realm of cardiac rehabilitation has been extensively discussed in the prior literature [57,58]. The reasons for dropout in this study align with those previously documented [59]. Most patients discontinued participation in the rehabilitation program due to financial constraints. The health insurance fund ceased subsidizing services, making payments challenging for them. Additionally, changes in personal circumstances, such as relocation, the birth of a child or grandchild, etc., were contributing factors. A small minority did not attend follow-up tests, and thus, their data were not included in the final analysis.

### 4.4. Implications for Cardiological Rehabilitation

The findings of this study have substantial implications for the field of CR. The advantages associated with heightened muscle strength extend beyond mere enhancements in physical capacity, encompassing improved overall well-being, enhanced quality of life, and diminished risk of subsequent cardiac events. The integration of strength training as a core component of remote rehabilitation introduces novel avenues for optimizing patient outcomes. The effectiveness of the program is not limited to improving compliance with strength training protocols; it also contributes to heightened patient engagement across the entire rehabilitation process.

Moreover, by embracing digital applications for remote monitoring and guidance, we have demonstrated the potential for scalability and accessibility in such programs. This holds particular significance in addressing the constraints associated with center-based rehabilitation.

### 4.5. Limitations and Future Directions

It is important to acknowledge some limitations of our study. The sample size was relatively modest and predominantly comprised low-risk male patients. Furthermore, the follow-up period was relatively short, aligning somewhat organically with the duration of providing patients with complimentary rehabilitation services within the healthcare system. Additionally, due to logistical challenges, we confined our study to a single rehabilitation center, thereby limiting the potential generalizability of our results.

Future research should consider larger, more diverse patient populations and long-term outcomes to further validate the effectiveness of the strength training program in cardiac rehabilitation.

Another notable limitation of our study was the substantial dropout rate. However, it is noteworthy that a 25% dropout rate is commonly observed in both general cardiac rehabilitation programs and comparable studies [60]. This dropout rate aligns with expectations considering the physical and emotional challenges associated with cardiac rehabilitation. Consequently, in anticipation of this potential dropout challenge, we initially enrolled participants at a 40% surplus compared to the minimum required for achieving the necessary statistical power.

## 5. Conclusions

In conclusion, despite all the limitations, our study introduces a novel approach to cardiac rehabilitation by emphasizing strength training in a remote setting. The promising results in terms of enhanced muscle strength, patient adherence, and satisfaction underscore the potential benefits of incorporating strength training in cardiac rehabilitation programs. This innovation represents a step forward in the endeavor to provide comprehensive care and support to cardiac patients, ultimately improving their overall health and well-being. Further research and continued development of such programs will contribute to a more holistic approach to cardiac rehabilitation.

## Figures and Tables

**Figure 1 jcm-13-01445-f001:**
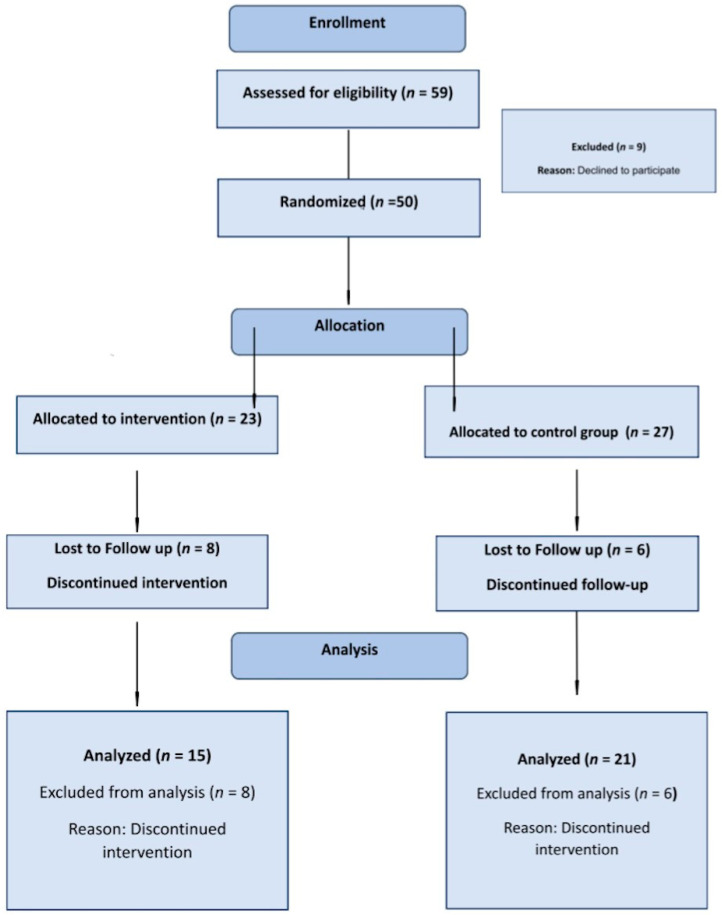
Consort 2010 Flow diagram.

**Figure 2 jcm-13-01445-f002:**
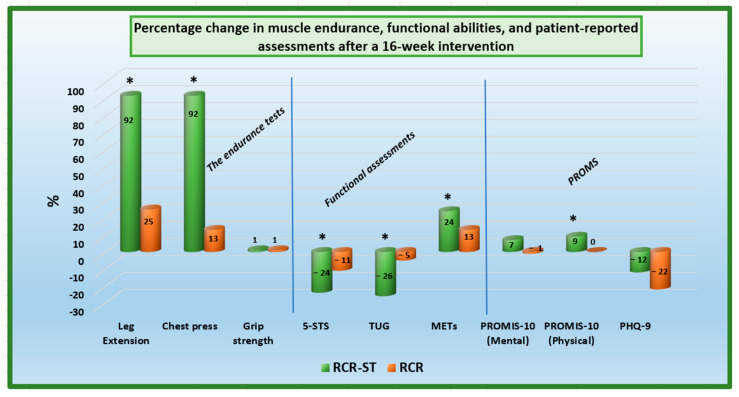
Percentage change in muscle endurance, functional abilities, and patient-reported assessments after a 16-week intervention, * *p* < 0.001.

**Table 1 jcm-13-01445-t001:** Demographics and clinical characteristics.

Variables	Intervention Group (*n* = 23)	Control Group (*n* = 27)	*p* Value
Age, years	57.7 ± 10.3	61.6 ± 10.3	0.189
Sex, male	18 (78.3)	26 (96.3)	0.050
Body metrics			
Weight, kg	83.2 ± 10.0	85.5 ± 17.8	0.577
Height, cm	174.2 ± 8.7	176.0 ± 9.6	0.479
BMI, kg/m^2^	27.4 ± 2.9	27.4 ± 4.5	0.980
Current Smoker	4 (17.4)	1 (3.7)	0.108
Past Smoker	6 (26.1)	3 (11.1)	0.170
Comorbidities			
HTN	8 (34.8)	7 (25.9)	0.496
Dyslipidemia	6 (26.1)	10 (37.0)	0.408
DM	3 (13.0)	4 (14.8)	0.857
CVA	1 (4.3)	1 (3.7)	0.908
Main Indication for CR			
IHD	17 (73.9)	23 (85.2)	0.321
MI-ACS	14 (60.9)	15 (55.6)	0.704
PCI	14 (60.9)	15 (55.6)	0.704
Valve Surgery	4 (17.4)	3 (11.1)	0.524
CABG	1 (4.3)	3 (11.1)	0.380
AFib	0	3 (11.1)	0.099
Chest Pain	6 (26.1)	7 (25.9)	0.990
Atrial Arrhythmias	1 (4.3)	4 (14.8)	0.219
STEMI	3 (13.0)	1 (3.7)	0.225
NSTEMI	5 (21.7)	7 (25.9)	0.730

Abbreviations: AFib, Atrial fibrillation; BMI, body mass index; CABG, coronary artery bypass graft; CR, cardiac rehabilitation; CVA, cerebrovascular accident; DM, diabetes mellitus; HTN, hypertension; IHD, ischemic heart disease; MI-ACS, myocardial infarction/acute coronary syndrome; PCI, percutaneous coronary intervention; STEMI, ST-elevation myocardial infarction; NSTEMI, Non-ST-elevation myocardial infarction. Data are presented as mean ± SD; or *n* (%).

**Table 2 jcm-13-01445-t002:** Repeated measures analysis of variance results for the time, group, and interaction effect between time and group of RCR-ST program.

	Intervention (*n* = 15)		Control (*n* = 21)		Time Effect	Group Effect	Time × Group Effect
	Baseline	16 Weeks	Baseline	16 Weeks	P (η2)	P (η2)	P (η2)
**Muscle Endurance**							
Leg Extension	15.4 ± 8.0	29.7 ± 14.3	18.6 ± 7.3	23.4 ± 13.0	<0.001 (0.515)	ns	0.005 (0.210)
Chest press	10.5 ± 7.1	20.2 ± 10.9	12.4 ± 9.6	14.1 ± 9.8	<0.001 (0.626)	ns	<0.001 (0.456)
Grip strength	36.8 ± 10.0	37.3 ± 10.5	42.8 ± 11.6	43.3 ± 12.0	ns	ns	ns
**Functional Assessments**							
5-STS (s)	10.7 ± 2.9	8.1 ± 2.1	10.7 ± 2.7	9.5 ± 3.0	<0.001 (0.513)	ns	0.021 (0.148)
TUG (s)	10.1 ± 2.5	7.4 ± 1.7	9.2 ± 1.6	8.7 ± 1.6	<0.001 (0.267)	ns	0.019 (0.152)
**Cardiorespiratory** **fitness**							
METs	8.5 ± 3.0	10.6 ± 3.4	9.7 ± 2.8	11.0 ± 2.6	<0.001 (0.591)	ns	ns

**Table 3 jcm-13-01445-t003:** Compliance and adherence scores of participants in the intervention group and the control group, bolded values highlight the statistically significant differences between the groups.

	Intervention (*n* = 15)	Control Group (*n* = 27)	*p* Value
Total minutes of weekly aerobic exercise	184.6 ± 113.3	229.1 ± 126.01	0.284
Minutes in the target heart rate per week	118.1 ± 100.5	95.9 ± 54.4	0.398
Minutes below the target heart rate per week	66.5 ± 39.0	133.2 ± 123.0	**0.051**
Borg scale per week	11.3 ± 1.8	10.6 ± 2.0	0.346
Number of weekly aerobic training sessions	5.8 ± 3.7	5.3 ± 2.1	0.614
Number of weekly resistance training sessions	2.0 ± 2.2	0.6 ± 0.7	**0.009**
Number of daily steps	7513.2 ± 1886.4	8298.6 ± 3194.0	0.362
Number of weekly entries in the mobile app	5.0 ± 1.1	3.7 ± 2.0	**0.012**
Duration (weeks) of participation in the CR program	15.1 ± 1.2	13.2 ± 3.9	**0.046**
Scores are means between 16 weekly measures			

**Table 4 jcm-13-01445-t004:** Pearson correlations between study variables at baseline.

		1	2	3	4	5	6	7	8	9
1	Leg Extension									
2	Chest press	0.59 ***								
3	Grip strength	0.25	0.32							
4	5-STS (s)	−0.45 **	−0.56 ***	−0.12						
5	TUG (s)	−0.39 *	−0.23	−0.25	0.44 **					
6	METs	0.56 **	0.70 ***	0.31	−0.60 ***	−0.38 *				
7	PROMIS-10 (Mental)	0.05	0.12	0.26	0.10	−0.36	0.13			
8	PROMIS-10 (Physical)	0.40 *	0.60 ***	0.21	−0.35	−0.39 *	0.53 **	0.54 **		
9	PHQ9	−0.47 *	−0.29	0.04	0.38	0.35	−0.35	−0.31	−0.62 **	

*** *p* < 0.001; ** *p* < 0.01; * *p* < 0.05.

## Data Availability

Privacy requirements and institutional review board obligations do not permit the publication of the raw data. Any reasonable requests should be addressed to the primary author.

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
