# Peer review of "Evaluating the Effects of an Enhanced Strength Training Program in Remote Cardiological Rehabilitation: A Shift from Aerobic Dominance—A Pilot Randomized Controlled Trial"

_jcm, 2024, doi:10.3390/jcm13051445_

Round 1
Reviewer 1 Report
Comments and Suggestions for Authors
The Manuscript title ‘’Evaluating the Effects of an Enhanced Strength Training Program in Remote Cardiological Rehabilitation: A Shift from Aerobic 3 Dominance - A Pilot Randomized Controlled Trial” by Irene Nabutovsky, ….. Mickey Scheinowitz is generally good and interesting and effectively addresses incorporating strength training into remote cardiac rehabilitation. The transition from emphasizing aerobic exercise to incorporating specific resistance training is innovative and highly applicable to present-day rehabilitation methods. The notable findings about muscular endurance and functional enhancements associated with increased patient involvement are very notable and provide essential knowledge to the discipline.
I appreciate the authors' organized conduction of this well-planned study and comprehensive examination of the results. The effective use of digital instruments for distant rehabilitation and emphasis on patient compliance and fulfillment are notable aspects of this study.
With minor modifications, I suggest considering the following overarching aspects:
1. The study acknowledges the dropout rates, but a more thorough investigation into the causes of these rates and their potential influence on the study's results would offer a more comprehensive comprehension of the difficulties encountered in remote rehabilitation settings.
2. The study highlights the strength training program's notable immediate advantages, without considering its potential long-term effects. The authors should consider addressing the long-term sustainability and monitoring of these effects, extending beyond the immediate timeframe of the study.
3. The study population is predominantly males and individuals with low-risk characteristics regarding diversity and inclusivity. Subsequent investigations should incorporate a more heterogeneous cohort of participants to ensure the generalizability of the results to a broader demographic.
Author Response
Dear Reviewer,
We have thoroughly reviewed all the comments and have made the necessary revisions and additions to our article. Enclosed, please find our responses addressing each of the points you raised.
Thank you for your attention to our manuscript.
Sincerely,
Irene Nabutovsky

Reviewer 2 Report
Comments and Suggestions for Authors
I am grateful to the editor for the opportunity to review the manuscript by Nabutovsky et al, "Evaluating the Effects of an Enhanced Strength Training Program in Remote Cardiological Rehabilitation: A Shift from Aerobic Dominance - A Pilot Randomized Controlled Trial." In this study, the authors examined the feasibility and effectiveness of using strength training in a remote cardiac rehabilitation program. This idea is novel and could potentially have important clinical implications. Until now, during remote cardiac rehabilitation, mainly aerobic training was used; the authors for the first time decided to use remote strength training, which determines the novelty of the study.
However, the scientific value of the presented data is questionable; the authors need to clarify the following issues:
1. In the Introduction section, it is necessary to clearly formulate the purpose of the study.
2. The article does not describe the method of randomizing patients. It is unclear why the groups ended up being unequal in the number of patients included in the study (23 and 27). There may have been an inclusion bias effect when patients were included in the study or control group.
3. The study had a large percentage of patients who dropped out; this could additionally affect the comparability of the groups for a number of indicators, which already had noticeable differences in the initial state, although they did not reach statistical significance. Therefore, it is necessary to provide data on the comparability of the groups included in the final analysis (15 and 21 patients).
4. It is necessary to clarify the reasons for leaving the study. Why did patients give up strength training? Inadequate program? Poor exercise tolerance?
5. In the Statistical analysis section, the authors did not provide data on whether the data were checked for normality of distribution. Judging by the small size of the groups, it is most likely that the distribution of quantitative data differs from normal. As a result, a different format for presenting data (median and quartiles), other programs for comparing data in groups and correlation analysis are required.
There is no information on how the authors determined the required power of the study.
6. In the main group, the initial indicators of muscle status were lower than in the control group. It is known that in patients with a low initial functional status, its increase is usually more pronounced. To what extent was this effect expressed in the present study?
7. The discussion section should begin with the main result obtained in the study.
Minor:
- the list of references is formatted incorrectly. When referring to publications in journals, there is no need to point to the INTERNET address of the journal page; the output data of the article is sufficient.
- on line 450 there is a link to source 60, but there is no such source in the list of references
Comments on the Quality of English LanguageNo comments
Author Response

(The authors gave the same response as above.)

Round 2
Reviewer 2 Report
Comments and Suggestions for Authors
The authors answered my questions and comments, but not all answers satisfied me.
1. The purpose of the study is usually placed at the end of the Introduction section. I see no reason for the authors to deviate from this tradition.
Minor - I did not see any changes in the bibliography
Comments on the Quality of English LanguageNo comments
Author Response
Dear Reviewer,
We have thoroughly reviewed all the comments and have made the necessary revisions and additions to our article. Enclosed, please find our responses addressing each of the points you raised.
Thank you for your attention to our manuscript.
Sincerely,
Irene Nabutovsky
- The purpose of the study is usually placed at the end of the Introduction section. I see no reason for the authors to deviate from this tradition.
Answer: Done. (page 2-3, lines 71-75)
- Minor - I did not see any changes in the bibliography
Answer: You are correct! Regrettably, the changes in the bibliography were not saved due to an oversight on our part. We sincerely apologize for this error. Rest assured, we have now adjusted the list of references to comply with the journal's requirements and removed any unnecessary links.
